# Isolation, Purification, and Characterisation of a Phage Tail-Like Bacteriocin from the Insect Pathogenic Bacterium *Brevibacillus laterosporus*

**DOI:** 10.3390/biom12081154

**Published:** 2022-08-20

**Authors:** Tauseef K. Babar, Travis R. Glare, John G. Hampton, Mark R. H. Hurst, Josefina O. Narciso

**Affiliations:** 1Bio-Protection Research Centre, Lincoln University, Lincoln 7674, New Zealand; 2Department of Entomology, Faculty of Agriculture Sciences & Technology, Bahauddin Zakariya University, Multan 60000, Pakistan; 3Resilient Agriculture, AgResearch, Lincoln Research Centre, Christchurch 8140, New Zealand

**Keywords:** antibacterial protein, contractile phage tail-sheath, defective phage, insect pathogenic bacterium, PBSX

## Abstract

The Gram-positive and spore-forming bacterium *Brevibacillus laterosporus* (*Bl*) belongs to the *Brevibacillus brevis* phylogenetic cluster. Isolates of the species have demonstrated pesticidal potency against a wide range of invertebrate pests and plant diseases. Two New Zealand isolates, *Bl* 1821L and *Bl* 1951, are under development as biopesticides for control of diamondback moth and other pests. However, due to the often-restricted growth of these endemic isolates, production can be an issue. Based on the previous work, it was hypothesised that the putative phages might be involved. During investigations of the cause of the disrupted growth, electron micrographs of crude lysate of *Bl* 1821L showed the presence of phages’ tail-like structures. A soft agar overlay method with PEG 8000 precipitation was used to differentiate between the antagonistic activity of the putative phage and phage tail-like structures (bacteriocins). Assay tests authenticated the absence of putative phage activity. Using the same method, broad-spectrum antibacterial activity of *Bl* 1821L lysate against several Gram-positive bacteria was found. SDS-PAGE of sucrose density gradient purified and 10 kD MWCO concentrated lysate showed a prominent protein band of ~48 kD, and transmission electron microscopy revealed the presence of polysheath-like structures. N-terminal sequencing of the ~48 kD protein mapped to a gene with weak predicted amino acid homology to a *Bacillus* PBSX phage-like element *xkdK*, the translated product of which shared >90% amino acid similarity to the phage tail-sheath protein of another *Bl* published genome, LMG15441. Bioinformatic analysis also identified an *xkdK* homolog in the *Bl* 1951 genome. However, genome comparison of the region around the *xkdK* gene between *Bl* 1821L and *Bl* 1951 found differences including two glycine rich protein encoding genes which contain imperfect repeats (1700 bp) in *Bl* 1951, while a putative phage region resides in the analogous *Bl* 1821L region. Although comparative analysis of the genomic organisation of *Bl* 1821L and *Bl* 1951 PBSX-like region with the defective phages PBSX, PBSZ, and PBP 180 of *Bacillus subtilis* isolates 168 and W23, and *Bacillus* phage PBP180 revealed low amino acids similarity, the genes encode similar functional proteins in similar arrangements, including phage tail-sheath (XkdK), tail (XkdO), holin (XhlB), and N-acetylmuramoyl-l-alanine (XlyA). AMPA analysis identified a bactericidal stretch of 13 amino acids in the ~48 kD sequenced protein of *Bl* 1821L. Antagonistic activity of the purified ~48 kD phage tail-like protein in the assays differed remarkably from the crude lysate by causing a decrease of 34.2% in the number of viable cells of *Bl* 1951, 18 h after treatment as compared to the control. Overall, the identified inducible phage tail-like particle is likely to have implications for the in vitro growth of the insect pathogenic isolate *Bl* 1821L.

## 1. Introduction

*Brevibacillus laterosporus* (*Bl*) is a Gram-positive and spore-forming bacterium belonging to the *Brevibacillus brevis* phylogenetic cluster in the family Paenibacillaceae [1]. Globally, various strains of this bacterium have exhibited biocontrol potential against invertebrate pests of different insect orders, including Coleoptera [2,3], Diptera [4,5,6], Lepidoptera [7], and also against nematodes [8] and molluscs [9,10]. The insecticidal action is mostly related to the production of diverse toxins, many of which act post-ingestion in the insect gut [11,12]. Due to the entomocidal action of *Bl* against diverse insect pests, there has been a surge of interest in using this species in microbial pesticides [13,14,15,16,17]. The New Zealand insect pathogenic isolates *Bl* 1821L and *Bl* 1951, originally isolated from brassica seeds, exhibited activity against diamondback moth (*Plutella xylostella*) and mosquito larvae (*Culex pervigilans* and *Opifex fuscus*) [7,11]. These isolates are under development as biopesticides, but often experience stymied growth during culturing, and it was hypothesised that bacteriophage might be involved [18]. Bacteria predominantly harbour prophages in their chromosomes [19], either in true or defective lysogenic forms [20,21]. The common products of defective phage assembly are sheath-like tails with activity against susceptible bacterial species [22,23]. Phage tail-like particles are commonly known as tailocins, defective bacteriophages, or cryptic phages [24,25] due to their potency against sensitive bacteria without injecting DNA [26]. Tailocins or phage tail-like bacteriocins (PTLBs) are bactericidal structures [25] first identified as R-type and F-type pyocins produced by *Pseudomonas aeruginosa* [27]. They resemble phage tails, with the R-type pyocins corresponding to the contractile tails of myophages such as T4 and the F-type pyocins corresponding to the flexible, non-contractile tails of siphophages such as T1 [27]. The common feature of the two forms is how they perpetuate in nature [28]. The best-studied examples are the colicins produced by *Escherichia coli* and the R-type pyocins of *P. aeruginosa*. Colicin gene clusters are encoded on plasmids while pyocins are typically located on chromosomal DNA [26,29]. Tailocins rely on receptor-binding proteins (RBPs) located on tail fibres or spikes for initial and specific interaction with susceptible bacteria [30,31]. Phages kill bacteria through a lytic, replicative cycle, whereas PTLBs kill the target cell through membrane depolarisation in a single hit mechanism [32]. 

Homologous bactericidal phage-like element PBSX protein defined within the Gram-positive bacteria of the genus *Bacillus* [33,34] is known to employ a similar killing mechanism to the one perpetrated by the tailocins [28,35]. PBSH, PBSX, PBSV, PBSW, PBSY, PBSZ, and PBP180 are some of the identified defective phages of *Bacillus subtilis* 168, *Bacillus licheniformis*, *B. subtilis* var. *vulgatus*, *B. subtilis* S31, *B. subtilis* W23, and *Bacillus pumilus* AB94180, respectively [33,36]. Among all of these bactericidal structures, PBSX from *B. subtilis* 168 has been the most widely studied phage-like element [34]. PBSX prophage is resident in the host chromosome and is inducible upon exposure to various DNA damaging agents such as mitomycin C, ultraviolet light, and hydrogen peroxide [37,38]. Induction releases synthesised structural proteins of phage-like particles ranging in mass from 12 kD to 76 kD [39,40], and is accompanied by the lysis of the cells [36]. Numerous phage tail-like structures with putative antibacterial activities have been identified in the Gram-negative bacteria [26] and in a limited number of Gram-positive bacteria, including *Listeria monocytogenes* [41], *Bacillus pumilus* [36], *B. subtilis* [42], *Clostridium difficile* [43], and *Bacillus aneurinolyticus* [44]. However, *B. pumilus* 15.1, is the only example of an entomopathogenic bacterium from which phage tail-like particle with bacteriocin activity was defined [45].

In this study we isolated, purified, and characterised the putative antibacterial phage tail-like structures of the PBSX-like region from an insect pathogenic isolate *Bl* 1821L and compared the encoding region to other bacteria, including the closely related strain *Bl* 1951.

## 2. Materials and Methods 

### 2.1. Bacterial Strains and Growth Conditions

The following bacterial strains were used in this study: *Bacillus megaterium* 3-2, *B. megaterium* S1, *B. subtilis* EM-13 (Tp5), *Bl* 1951, *Bl* 1821L, *Bl* Rsp, *Bl* CCEB 342, *Bl* NRS 590, *Bl* NCIMB, *Carnobacterium maltaromaticum* 3-1, *Fictibacillus rigui* EM-14 (FJAT 46895), *Oceanobacillus* sp. EM-12 (R-31213), and *Oerskovia enterophila* 3-3. All the strains are held at −80 °C in the Bio-Protection Research Centre Culture Collection, Lincoln University, New Zealand except *Bl* CCEB 342 and *Bl* NRS 590. The strains *Bl* CCEB 342 and *Bl* NRS 590 were kindly provided by Professor Colin Berry, Cardiff University, UK.

Luria-Bertani (LB, Miller) medium broth was routinely used for growing bacteria on an orbital shaker (Conco, TU 4540, Taibei, Taiwan) at 250 rpm and 30 °C overnight for further usage. 

### 2.2. Mitomycin C Induction of Putative Antibacterial Protein

Mitomycin C (Sigma, Sydney, NSW, Australia) was used to induce the putative antibacterial proteins as outlined by [46]. Single colonies of bacteria were used to inoculate 5 mL of LB (Miller, Sigma) broth, which was placed on an orbital shaker (Conco, TU 4540, Taibei, Taiwan) at 250 rpm and 30 °C overnight. Aliquots (500 µL) of the overnight culture were transferred to further inoculate 25 mL of LB broth. The culture was left to grow (~10–12 h) at 250 rpm and 30 °C until it attained turbidity. Two concentrations (1 µg/mL and 3 µg/mL) of mitomycin C (Sigma) were independently added into the flasks, which were then positioned on an orbital platform for incubation with rotation at 40 rpm at ambient temperature (24°C). OD_600nm_ readings were recorded through an Ultrospec-10 spectrophotometer (Amersham Biosciences, Amersham, UK) after 2, 4, 6, and 24 h of mitomycin C (Sigma) addition. The flasks were monitored for signs of cell lysis (clearing of the culture or accumulation of bacterial debris). The flasks without the addition of mitomycin C served as a control. All the treatments in the experiment were technically replicated four times. OD_600nm_ readings at various time intervals were pooled and statistically analysed using the ANOVA (Analysis of Variance) test through the Genstat 20th edition programme.

### 2.3. Soft-Agar Overlay Method with Polyethylene Glycol (PEG) Precipitation

Antibacterial activity of bacteriophages and phage tail-like structures (bacteriocins) can be differentiated after PEG 8000 precipitation of mitomycin C-induced bacterial cultures and the serial dilution assay testing in the soft-agar overlay [47]. For PEG precipitation, mitomycin C-induced cultures were centrifuged at 16,000× *g* for 10 min, and the supernatants were passed through a 0.22 µm filter. To the filtered cell free supernatant (CFS), 1M NaCl and 10% PEG 8000 were added and the sample was repeatedly inverted until both the NaCl and PEG 8000 were completely dissolved. The sample was incubated in an ice bath for 60 min and subsequently centrifuged (16,000× *g*, 30 min, 4 °C). The supernatant was decanted and the pellet was resuspended in 1/10th volume of the original supernatant volume of buffer (10 mM Tris, 10 mM MgSO4, pH 7.0) by repeated pipetting. PEG 8000 residue was removed by two sequential extractions with an equal volume of chloroform, which was combined with the resuspended pellet and vortexed for 10-15 sec. The mixture was centrifuged (16,000× *g*, 10 min), and the upper aqueous phase was transferred to a fresh 1.5 mL microfuge tube. This extraction process was repeated until no white interface between the aqueous and organic phases was visible. Soft agar (0.5%) was prepared and maintained at 55–60 °C before use. Next, 3 mL of soft agar was transferred into a 15 mL sterile tube and 100 µL of overnight culture was added. The tube was rotated for 10–15 s and then immediately poured over an LB agar plate. LB agar plates were allowed to dry for 20–30 min. Tenfold serial dilutions of PEG 8000 precipitated culture of *Bl* 1821L were prepared. Once the agar was solidified, 10 µL of PEG 8000 precipitated supernatant of each dilution (undiluted to 10^−8^) was pipetted at 2 spots on the lawns of host bacterium, and the plates were incubated at 30 °C for 48–72 h. The control comprised spotted sterile LB broth. Three LB agar plates were used for each dilution to find out the antibacterial activity of the PEG 8000 precipitated lysate. Antibacterial activity was evaluated by measuring the diameter of the zone of inhibition (mm) at the spotting point. 

### 2.4. Antimicrobial Spectrum of PEG 8000 Precipitated Culture Filtrates

To assess the antimicrobial spectrum of PEG 8000 precipitated *Bl* 1821L filtrate after mitomycin C induction against itself and other Gram-positive bacteria (*Bl* 1951, *Bl* Rsp, *Bl* CCEB 342, *Bl* NRS 590, *C. maltaromaticum* 3-1, *B. megaterium* 3-2, *B. megaterium* S1, *O. enterophila* 3-3, *Paenibacillus* sp. 15.12.1, *Oceanobacillus* sp. R-31213, *B. subtilis* Tp5, and *F. rigui* FJAT 46895), the soft agar overlay method was used as outlined above. Ten µL of sterile LB broth pipetted on the plates was used as a negative control. For the antimicrobial spectrum determination assay, three LB agar plates were used. Antimicrobial activity was evaluated by measuring the diameter (mm) of the zone of inhibition at the spotting point. 

### 2.5. Purification of Putative Antibacterial Protein

The crude lysate of *Bl* 1821L harbouring the putative antibacterial proteins after mitomycin C induction was purified by ultracentrifugation of 7.5 mL of CFS at 35,000 rpm (151,263× *g*) in a swing bucket rotor (41Ti, Beckman, California, USA) for 70 min. The concentrated pellet was resuspended in 100-150 µL of TBS buffer (40 g NaCl, 1 g KCl, 2.42 g TRIS Base, 16.5 g TRIS-HCl, dH_2_O 490 mL, pH 7.4 in 500 mL). Sucrose density gradients were formed by applying layers of 1.25 mL of freshly prepared sucrose solution 10%, 20%, 30%, 40%, 50%, and 60% sequentially from the highest to the lowest concentrations. Two hundred μL of the ultracentrifuged preparation was applied on top of the gradients from where the samples were centrifuged at 35,000 rpm (151,263 × *g*) in a swing bucket rotor (41Ti, Beckman) for 70 min. 

After ultracentrifugation, individual gradients were assayed against *Bl* 1821L and *Bl* 1951 as the host bacterium using the Kirby–Bauer disc diffusion test [48]. After the disc diffusion assay, each of the gradients was ultracentrifuged to concentrate the pellet. The purified and concentrated solution of 60% sucrose density gradient pellet was then cleaned using an Amicon Ultra-0.5 (10 kD) centrifugal filter (Millipore, Cork, Ireland) for sodium dodecyl sulphate polyacrylamide gel electrophoresis (SDS-PAGE) and transmission electron microscope (TEM) analysis.

### 2.6. TEM Examination of Crude and Purified Lysates

For TEM analysis of crude lysate of *Bl* 1821L, 7 mL of CFS of *Bl* 1821L from mitomycin C-induced culture was ultracentrifuged at 35,000 rpm for 70 min and 4 °C. The concentrated pellet was dissolved in 150 µL of 25 mM TBS. A 5 µL sample was applied to a freshly glow-discharged plastic-coated hydrophilic 200 mesh EM grid (ProSciTech; Thuringowa, Australia) and stained with 3 µL of 0.7% uranyl acetate (UA, pH 5). The *Bl* 1821L sample was subjected to TEM analysis at 18,000 to 25,000 magnification in Morgagni 268D (FEI, Hillsboro, OR, USA) TEM operated at 80 KeV. The images were photographed using the TENGRA camera. 

### 2.7. SDS-PAGE Analysis 

The putative antibacterial proteins of *Bl* 1821L were subjected to SDS-PAGE analysis using 30% acrylamide [49]. Next, 10 µL of protein ladder (BIO-RAD, Precision Plus Protein^TM^ Standards) was loaded. Gels were run for 50 min at 200 volts and then washed four times with H_2_O before staining with RAMA stain (29 mL MQW, 12.5 mL CBB stock (1 gm CBB, 300 MeOH, 200 mL MQW), 3.75 mL ammonium sulphate (200 gm, 500 mL MQW), 5 mL of glacial acetic acid per gel) [50]. Prior to overnight destaining in water, the gel was rinsed in water. However, for silver staining, the protocol of [51] was followed.

### 2.8. Bactericidal Activity Assay 

First, 500 µL crude lysate harbouring the putative antibacterial proteins (ABPs) was pipetted into the overnight culture of the host bacteria. For the control treatment, a similar volume of LB broth was used. All the flasks with/without putative ABPs were maintained at 30 °C on a shaking incubator (Conco, TU 4540, Taibei, Taiwan). Samples were drawn from each treatment after 1, 3, 6, 12, 18, and 24 h of incubation to determine the number of viable cells (CFU/mL) and OD_600m_. Cell biomass (OD_600nm_) was measured using the Ultrospec-10 spectrophotometer (Amersham Biosciences). To determine the number of viable cells (CFU/mL), tenfold serial dilutions (10^−1^ to 10^−6^) of each time interval were prepared and 100 µL from each dilution was spread in duplicate on two independent LB agar plates. After incubation at 30 °C for 2–3 days, the colonies were counted using a colony counter (Stuart, London, UK) and converted into log_10_ CFU/mL. To define potential ABPs antagonistic activity, the percentage change in number of viable cells compared to the control treatment (without ABPs) after treatment with the putative ABPs was calculated. Four independent set of experiments with biological replications were performed, and the pooled data were subjected to statistical analysis using the ANOVA (Analysis of Variance) test through the Genstat 20th edition programme. Similar to the above, an experiment with the purified PTLP was performed.

### 2.9. N-Terminal Sequencing and Bioinformatic Analysis 

A prominent band (~48 kD) of the filtered and concentrated supernatant (crude) of *Bl* 1821L on the SDS-PAGE was subjected to N-terminal sequencing. After the initial assessment, the purified gel bands of the protein of interest from SDS-PAGE were excised for N-terminal sequence analysis. Peptides generated by trypsin (Promega, Madison, WI, USA) cleavage from separated protein bands of interest were analysed using liquid chromatography mass spectrometry (LC-MS) for further identification. The obtained peptides were compared with the predicted proteins from the *Bl* 1821L genome (NZ_CP033 464.1). Amino acid sequences from the genomes were assessed and characterised using Uniprot database (https://www.uniprot.org; accessed on 25 September 2019), BLASTp (Basic Local Alignment Search Tool) (https://blast.ncbi.nlm.nih.gov; accessed on 23 April 2021), ExPasy (https://www.expasy.org; accessed on 23 April 2021), and the CLUSTALO (https://www.uniprot.org; accessed on 23 April 2021).

The identified ~48 kD putative antibacterial protein of *Bl* 1821L was analysed using Geneious [52]. Amino acid sequence alignments were performed using CLUSTALO (https://www.uniprot.org; accessed on 23 April 2021). AMPA (Antimicrobial Sequence Scanning System) is a bioinformatic tool that identifies the antimicrobial domains of proteins (http://tcoffee.crg.cat/apps/ampa; accessed on 23 April 2021) [53]. We ran the whole protein sequence in AMPA using a default propensity value of 0.225. Amino acids expressing the propensity value below the threshold level were considered a positive hit for latent bactericidal activity.

## 3. Results

### 3.1. Mitomycin C Induction of Putative Antibacterial Proteins

OD_600nm_ reading of the *Bl* 1821L culture after treatment with the mitomycin C at a concentration of 1 µg/mL or 3 µg/mL significantly declined by 49.7% and 41% after 24 h, respectively. The OD_600nm_ reading of the control treatment (without mitomycin C) revealed a slight increase in cells up to 6 h from where the OD_600nm_ slowly declined from 6–24 h post incubation (hpi) (Figure 1 and Appendix A). 

### 3.2. Antimicrobial Activity of PEG 8000 Precipitated Culture Filtrates

Assay testing of the PEG 8000 precipitated filtrates in the serial dilutions in our work indicted an increase in turbidity (Appendix A), but this antagonistic activity was lost upon transfer to fresh soft-agar lawn, which is an indication of bacteriocinogenic activity. *Bl* 1821L PEG 8000 precipitated filtrates showed inhibitory activities against itself at the highest concentration (FS—full strength, undiluted) (Table 1 and Appendix A) and against *Bl* 1951 between FS and 10^−3^ dilution (Table 1 and Appendix A). 

Using 1 µg/mL of mitomycin C induced culture of *Bl* 1821L after PEG 8000 precipitation exhibited high activity against all the tested strains of *Bl* except against itself and *Bl* CCEB 342 (Table 2). With the exception of *C. maltaromaticum* isolate 3-1, which was slightly sensitive to inhibitory action, all other tested Gram-positive bacterial strains were not susceptible to the *Bl* 1821L PEG 8000 precipitated culture supernatant (Table 2 and Appendix A). 

### 3.3. TEM and SDS-PAGE Analysis

Electron micrographs of concentrated crude lysate of *Bl* 1821L showed the presence of incomplete phage structures similar to hollow sheath and contractile tail sheath-like structures as well as the phage capsid-like and cog wheel-like structures (Figure 2B,C). TEM examination of purified and 10 kD MWCO membrane concentrated lysate revealed the presence of polysheath-like structures (Figure 2D,E) which were jointed at some points, with (Figure 2D) or without (Figure 2E) knot-like structures. 

A prominent band of ~48 kD was visualised after high-speed centrifugation of the crude lysate on SDS-PAGE (Appendix A) and it was hypothesised to be involved in the putative antibacterial activity. Furthermore, SDS-PAGE of sucrose density gradient (60%) purified and 10 kD MWCO membrane concentrated *Bl* 1821L lysate authenticated the preliminary findings by showing a prominent band of ~48 kD (Figure 2A).

### 3.4. Identification of a ~48 kD Putative Antibacterial Protein in Bl 1821L Genome

Alignment of the ~48 kD purified protein N-terminal sequences revealed several hits (covering approximately 34% of the amino acid sequence; Appendix A) to the loci A0A518VEB0 in the *Bl* 1821L genome (NZ_CP033464.1) encoding a predicted defective phage protein, similar to that encoded by the *Bs* 168 phage-like element PBSX gene *xkdK*. Through Uniprot, the predicted ~48 kD XkdK-like protein of *Bl* 1821L shared high amino acid identity with phage tail, phage tail sheath, and uncharacterised proteins (Appendix A). BLASTp analysis of the identified accession (A0A518VEB0) against Uniprot database exhibited 90.3% amino acid similarity to the phage tail-sheath protein of *Bl* LMG 15441 and a phage tail protein, phage sheath protein, and uncharacterised proteins encoded in various *Bl* strains (Table 3). Furthermore, these findings validated the preliminary results of the N-terminal sequencing of the prominent band (~48 kD) of the SDS-PAGE (Appendix A). 

### 3.5. Bactericidal Activity of Identified XkdK Protein

The crude lysate containing the identified XkdK protein along with the other putative ABPs after 6 h of incubation at 30 °C caused a non-significant decrease of 30.1% in the number of viable cells of *Bl* 1821L compared to the control (without ABPs) and preceding time intervals (1 and 3 h) in the liquid assay but showed no effect against *Bl* 1951 (Appendix A). 

AMPA analysis of the *Bl* 1821L phage-like element PBSX protein XkdK identified a bactericidal stretch (Appendix A) which was found from 360 to 373 amino acids (NKKFAKNIVRVLD) (Appendix A, shown in red) with a propensity value of 0.223 for the sequenced phage tail-like protein of *Bl* 1821L. In contrast to the activity of crude lysate, the addition of purified ~48 kD phage tail-like protein (PTLP) caused a decline of 22.1% and 34.2% after 12 and 18 h in the number of viable cells of the *Bl* 1951 compared to the control (without PTLP) (Appendix A and Figure 3). Similar to crude lysate, OD_600nm_ assessments of the treated and untreated cultures revealed no prominent differences across the assessed time points (Appendix A).

### 3.6. Bl 1821L and Bl 1951 Phage-Like Element PBSX Protein XkdK Comparison with the Similar Proteins of Other Gram-Positive Bacteria and Bl Phages

The predicted ~48 kD phage like-element PBSX protein *XkdK* of *Bl* 1821L and *Bl* 1951 was compared with identical XkdK proteins/phage tail-sheath proteins of other Gram-positive bacteria belonging to the genus *Aneurinibacillus*, *Bacillus*, *Brevibacillus*, *Clostridioides*, and *Geomicrobium* (Appendix A). Amino acid alignment of the *Bl* 1821L and *Bl* 1951 XkdK sequence with orthologous proteins provided an insight into their phylogenetic relationships, revealing mainly distant relationships (Appendix A and Figure 4). Despite similar organisation of the operons, the specific *Bl* 1821L XkdK protein had only 21.2% amino acid similarity with the *Bs* 168 XkdK protein (Appendix A and Figure 5 and Appendix A). The predicted phage tail-sheath protein XkdK of *Bl* 1821L and *Bl* 1951 aligned with the phage tail-sheath proteins of different known *Bl* phages, including Abouo, Davies, Jimmer2, Powder, and Osiris [54], but only shared 21.9% amino acid similarity (Appendix A). A dendrogram of aligned phage tail sheath proteins of *Bl* defined phages with the similar protein of *Bl* 1821L and *Bl* 1951 also indicated a distant relationship (Appendix A and Figure 6). Furthermore, this analysis was supported through CLUSTALO-based amino acids alignment (Appendix A).

### 3.7. Bioinformatic Analysis of Regions Adjacent to the Bl 1821L and Bl 1951 Putative Antibacterial Protein Encoding Gene 

The identified homolog of the phage-like element PBSX gene *xkdK* in the *Bl* 1821L genome (NZ_CP033464.1) is encoded between 5,137,125 bp and 5,138,462 bp. Bioinformatic analysis of genomic regions flanking the *xkdK* gene (A0A518VEB0) in both *Bl* 1821L and *Bl* 1951 revealed some genetic similarities to the PBSX-like region described in *Bs* 168 (Appendix A). An ortholog of the *Bl* 1821L phage tail-sheath encoding gene *xkdK* was identified in the genome of *Bl* 1951 (RHPK01000003.1, contig 1; 1,937,150 bp to 1,935,813 bp). In *Bl* 1951, at the 5′ end of the *xkdK* gene were two ORFs encoding hypothetical proteins (Appendix A); the one immediately upstream was identical to an uncharacterised protein (A0A502IL32). The adjacent upstream hypothetical protein (A0A502IL09) shared 96% amino acid similarity with the XkdN protein of *Bl* LMG 15441 (A0A075R977). Furthermore, in the *Bl* 1951 genome, several other PBSX-like gene orthologues such as *xkdT* and *xkdU*, as well as other phage and hypothetical genes are present (Appendix A). 

Bioinformatic analysis comparing the PBSX-like regions of *Bl* 1821L and *Bl* 1951 revealed that *Bl* 1951 had two glycine rich protein encoding genes with 1700 bp imperfect repeats in one region, while a predicted unnamed phage resides in the analogous region in *Bl* 1821L genome (Figure 7).

Furthermore, analysis of this genomic region in both the *Bl* 1821L and *Bl* 1951 identified genes, the translated products of which are involved in phage lysis activities, including N-acetylmuramoyl-l-alanine amidase and holins. An upstream gene encoding a protein homologous to a putative bacteriocin UviB (BhlA) of *Bs* 168 along with homologs of other phage-like element PBSX genes *xkdT* and *xkdU* also resided in *Bl* 1821L PBSX-like region (Figure 8 and Appendix A). BLASTp analysis of the holin protein (BhlA) exhibited homology to an uncharacterised protein of *Bl* LMG 15441, with protein orthologues in several *Bl* phages (Appendix A). A hydrolytic enzyme, N-acetylmuramoyl-l-alanine amidase, a regulating gene, is adjacent to the *uviB*/holin *bhlA* (Appendix A). Furthermore, several other phage-like proteins including a tail protein, phage FluMu protein Gp47, and a prophage LambdaBa01 Xpf encoded in the PBSX-like region of *Bl* 1821L are also annotated. Various genes encoding hypothetical proteins are also localised in the region where *xkdK* resides in the *Bl* 1821L genome (Figure 8). Located at the 3′ end of the phage-like element PBSX in *Bl* 1821L are four (ABC transporter) permease genes *yvcR*, *yvcS*, *yvcQ*, and *yvcP* (Appendix A). The translated products of these genes have been implicated in the export of lipid II-binding lantibiotics, such as nisin and gallidermin. 

### 3.8. Genomic Comparison of the Bl 1821L and Bl 1951 PBSX-Like Region with the Defective Prophages PBSZ, PBSX, and PBP180 

Comparison of the genomic organisation of PBSX-like regions of *Bl* 1821L and *Bl* 1951 with the analogous regions of defective prophages PBSZ, PBSX, and PBP180 of *B. subtilis* W23, *B. subtilis* 168, and *Bacillus* phage PBP180 revealed similar architecture (Figure 8). However, despite organisational similarities, significant differences were noted. Located at the 5′ end of *Bl* 1821L *xkdK* are two terminase unit (*xtmB*) and *xkdN* genes, while at the 3′ end are genes encoding uncharacterised proteins, a prophage LambdaBa01, and positive control factor Xpf reside (Figure 8). Immediately upstream of the *Bl* 1821L *xkdN* gene encoding is the tail protein corresponding to *xkdO* gene of PBSX, PBSZ, and PBP180 genome; but in other *Bacillus* strains, a gene *xkzB* encoding an uncharacterised protein is located at the 3′ of the tail protein gene *xkdO*. The gene encoding XkdP, a murein-binding protein, resides 5′ of the phage tail protein of the *Bacillus* strains, and several genes encoding the hypothetical proteins (XkdQ, XkdR, XkdS) reside upstream of it (Figure 8). 

Similar to *Bl* 1821L, the encoded PBSX-like region in the *Bl* 1951 genome was analysed to compare the genomic organisation with that of PBSZ, PBSX, and PBP180 (Figure 8). Located 5′ of *xkdK* is a gene encoding a hypothetical protein. Within the PBSX-like region of *Bl* 1821L and *Bl* 1951, the section which differs (Figure 7) is notable in *Bacillus* strains due to the presence of genes encoding XkdV (a putative tail fibre protein) and hypothetical proteins (Figure 8). Amino acid alignment of the XkdK protein of *Bl* 1821L and *Bl* 1951 sharing 22.9%, 22.9%, and 21.6% amino acid identity exhibited low amino acid identity with the respective protein orthologues PBSZ, PBSX, and PBP180 of strains *B. subtilis* W23, *B. subtilis* 168, and *Bacillus* phage PBP 180 (Appendix A and Figure 9). 

The XkdO (A0A518VEA0) tail protein residing in the PBSX-like region of *Bl* 1821L and *Bl* 1951 shares 50% amino acid similarity with the corresponding protein of PBSZ prophage but shared no amino acid similarity to the PBSX and PBP180 defective prophage tail proteins. However, the PBSX-like region protein XkdO shared 51.1% and 37% amino acids similarity with the respective PBSZ and PBP180 tail proteins (Appendix A). 

The holin protein (XhlB) of PBSX shared 89.8% and 61.4% amino acid similarity with the respective PBSZ and PBP180 amino acid sequences. In contrast, the *Bl* 1821L and *Bl* 1951 holin protein (BhlA) exhibited a low level of 11.4%, 13.6%, and 13.6% amino acid identity with the PBP180, PBSZ, and PBSX (Appendix A). Distance matrices generated using Geneious basic (Appendix A) and amino acid alignment using CLUSTALO (Appendix A) substantiated these findings. N-acetylmuramoyl-l-alanine is an endolysin-related protein (XlyA) localised in the *Bl* 1821L and *Bl* 1951 genome, which had a low sequence similarity of 15.4%, 15.4%, and 7.7% to PBSZ, PBSX, and PBP180, respectively. However, these proteins were quite similar to each other; PBSX had amino acids similarity of 95.3% with PBSZ and 53.9% with PBP180 (Appendix A). 

## 4. Discussion

While deciphering the causes of erratic growth of entomopathogenic strains of *Bl*, cultures induced with mitomycin C showed a decline in cells biomass that correlated with the presence of bioactive antagonists in *Bl* 1821L PEG precipitates. Furthermore, serial dilution testing of precipitated filtrates and the electron micrographs of the crude lysates authenticated the absence of phage activity. N-terminal sequencing of a prominent ~48 kD protein in semi-purified bioactivity samples identified a phage-like element PBSX gene *xkdK*. However, antibacterial activity of the crude lysate harbouring different putative ABPs including the ~48 kD PTLP differed from the purified ones. AMPA analysis predicted the bactericidal potency of the purified protein. TEM analysis of the purified ~48 kD protein showed the presence of polysheath-like structures. In the liquid growth assay, a decrease in the number of viable cells of *Bl* 1951 was noted after treatment with the purified PTLP, which had genetic similarities to the phage-like element PBSX protein XkdK. In total, our results indicated that a phage tail-like element, encoded by a PBSX-like region of *Bl* 1821L and *Bl* 1951 isolates, was antibacterial (bacteriocin) in nature. 

Occasionally, the incomplete phage particles may not fit well with the description of R-type or F-type bacteriocins but could correspond to phage killer particles [21]. Interestingly, these elements behave like bacteriocins but are very diverse genetically, displaying features ranging from streamlined PTLBs to nearly complete phages [26]. Therefore, for the defective prophages the term “phage killer particles or protophages” was coined to include PBSX and other non-infectious defective phage particles acting as de facto bacteriocins [33,55]. Numerous defective phage-like elements including PBSX are inducible via the SOS response [36,38], and the induction is suicidal for the bacterial cells, as it results in lysis [56,57]. Turbidimetry is one of the established methods used to monitor bacterial growth since optical density (OD_600nm_) measurements make it possible to follow bacterial population growth in real time. Some authors have attempted to derive growth parameters from optical density measurements [58,59]. In our study, using mitomycin C at 1 µg/mL significantly decreased the cell biomass (OD_600nm_) of *Bl* 1821L by 49.7% after 24 h compared to the control. Following the induction, the lysate was examined under TEM and structures resembling to incomplete phage particles were observed. The findings were in agreement with the work of [45] on an insect pathogenic strain *B. pumilus* 15.1. Through this study, we have defined the concentration of mitomycin C for the induction of phage tail-like structures of *Bl* 1821L. Phage-induced bacterial lysis relies on the concerted action of two proteins (lytic enzymes), holin and endolysin [60,61]. Bioinformatic analysis identified a holin protein (BhlA) similar to that found in *Bs* prophage SPβ than the PBSX holin XhlA, a sign of genomic plasticity that phages and phage-related particles have [45]. The timing of lysis, which is critical for viral reproduction, is somehow “programmed” into the structure of the holin [62]. Localisation in the region of lysis genes holin (*bhlA*) and endolysin enzyme N-acetylmuramoyl-l-alanine amidase (*xlyA*) suggests that the predicted lysis genes of the *Bl* 1821L PBSX-like region may follow a pathway similar to the phages in host cell lysis. Hypothetical proteins resided at 3′ of the holin (BhlA) protein of *Bl* 1821 and *Bl* 1951 PBSX-like region. While analysing the genome of different *B. pumilus* strains [45], speculated that these hypothetical proteins might be the part of the lysis module, based on the location of the similar proteins near holins and endolysins. Typically, fully functional prophages tend to undergo a round of lytic growth, resulting in plaque formation, while a substantial number of phage-like entities present in bacterial genomes only produce phage tail-like structures [63]. Phage tail-like particles are devoid of the phage genome and cannot multiply and form plaques [47]. Therefore, in the present study, the soft-agar overlay method with PEG 8000 precipitation was used to differentiate between the activities of bacteriophages and bacteriocins [47]. Assay test findings were in accordance with previous work, which demonstrated that performing a serial dilution on a supernatant containing bacteriophages will result in individual plaques becoming less in number with greater dilution, whereas in the former it will result in a clearing zone that becomes uniformly more turbid with greater dilution [47,64]. Furthermore, a bacteriophage will produce a clearing zone when spotted onto a fresh soft-agar overlay seeded with the same strain, whereas a bacteriocin will not produce a clearing zone when transferred to a fresh soft agar lawn, owing to the dilution of the bacteriocin [47]. Using the same method, bacteriocin-like activity of the phage tail-like particles from different Gram-positive bacteria were defined [43,45].

Bacterial genomes often harbour genes encoding a toxin–antitoxin (TA) system [65]. Within this system, the toxin may inhibit the producing cell’s own cell growth by halting essential cellular processes [66], but its cognate antitoxins can block this activity [67]. In our study, the crude lysate harbouring the different putative ABPs including the ~48 kD PTLP, statistically caused a non-significant decrease of 30.1% in the number of viable cells of *Bl* 1821L and no effect against the *Bl* 1951. However, the addition of purified ~48 kD PTLP into the culture of *Bl* 1951 decreased the number of viable cells by 34.2%, 18 h after treatment, compared to the control (without PTLP). Variation in the bactericidal activity of crude lysate and the purified PTLP suggests that a TA system might be operational in a crude form that neutralises the lethal effect of one protein against the other. The killing mechanism of phage tail-sheath proteins (bacteriocins) is related to the mechanism in which *Myoviridae* phages translocate DNA into the cell after binding, but instead of DNA entering the cell, a pore is created [26,30]. These findings are in agreement with previous work, which showed that defective phages such as phage-like element PBSX containing the protein XkdK [45] and R-type tailocins [35] act as specialised weapons against kin bacteria. 

N-terminal sequence of ~48 kD purified protein band identified a gene with low homology (34%) to the phage-like element PBSX gene *xkdK* similar to that encoded in the *Bs* 168. PBSX of *Bs* 168, is one of the most widely studied defective phages [36]. The region flanking *xkdK* encoded a phage-like element and resembled a PBSX-like encoding region. The PBSX phage particle is composed of at least 26 polypeptides. XkdG is the main capsid protein; XkdK and XkdM are the tail sheath and core proteins, respectively. XkdV is the fibre subunit that is involved in the killing of susceptible strains [39,68]. In phages, long tails may be contractile (*Myoviridae*) or non-contractile (*Siphoviridae*), and the morphogenesis of the tail is largely dependent on the production of the tail assembly chaperon protein, which is XkdN in PBSX [69]. XtmB proteins (phage terminase large subunit) in other PBSX-like structures reside in the capsid region and often accompany the small terminase subunit protein XtmA. However, in the *Bl* 1821L putative PBSX-like region, their location in the upstream region suggests a possible role in the adhesion of tail protein with the putative phage tail-sheath protein. Temperate phages integrating into the genome of the susceptible host attach themselves with an enzyme known as integrase [70]. An integrase-encoding gene was found in the *Bl* 1821L region and is presumed to be a site of attachment of *xkdK* gene in the *Bl* 1821L genome.

Another New Zealand isolate, *Bl* 1951, which has been shown to harbour the specific phage-like element PBSX gene *xkdK*, exhibited a similar genomic organisation around the gene. However, PBSX-like regions in both the isolates differed with the insertion of 4458 bp at the 3′ end of *xkdT* gene in *Bl* 1821L compared to *Bl* 1951. In *Bl* 1951, in the upstream region of *xkdT* and *xkdU*, are genes encoding imperfect repeats within glycine rich proteins. The tips of phage tail fibres are reported to be rich in glycine [71,72]. Glycine-rich proteins are considered vital in sustaining the unusual conformations found in the tail fibre structures [73]. Based on this information, it is likely that the glycine-rich proteins of *Bl* 1951 might be involved in conformations of tail fibres for adhesion to its host and have a role in defining the antimicrobial spectrum. Phage tails are the molecular machines that play a decisive role in determining the host specificity and infection process of respective phages [73,74]. Tail fibres, tail spikes, and tail tips located at the distal end of tailed phages act as receptor binding proteins (RBPs) to interact with the bacterial cell surface receptors [75]. This initial recognition subsequently leads to infection of the bacterium by phages or the depolarisation of its membrane by PTLBs [30,76]. Typically, phages and PTLBs exhibit a narrow host range [77] and employ the same primary determinant of specificity, i.e., one or more RBPs used to bind to the cell surface of the sensitive host bacterium [74]. For instance, PBSX-like particles isolated from an entomopathogenic bacterium *B. pumilus* 15.1 in the spot assay test displayed putative antibacterial activity against the closely related strains (narrow spectrum) [45]. In this study, PEG 8000 precipitated lysate harbouring the putative phage tail-sheath protein exhibited broad-spectrum antagonistic activity that included the kin strains *Bl* 1951, *Bl* Rsp, *Bl* NRS 590, *Bl* NCIMB, and an unrelated species, *C. maltaromaticum*. The sensitivity of *Brevibacillus* strains and the other Gram-positive bacteria implied that these might share similar RBPs. These findings align with the previous studies where the putative antibacterial activity of R-type pyocins against other bacteria like *Campylobacter* sp. [78]; *Haemophilus influenzae* [79]; *Haemophilus ducreyi* [80]; *Neisseria gonorrhoeae* [81]; and *Neisseria meningitidis* [82] was attributed to the common receptor sites on *P. aeruginosa* [83]. There are several examples in the past where the host range of bacteriophages and PTLBs has been expanded due to modifications in the tail fibre proteins [32,84]. 

Comparative analysis of the genomic architecture of the *Bl* 1821L PBSX-like region with the defective prophages PBSZ, PBSX, and PBP180 demonstrated some shared features, while showing low homology for individual genes. Despite sharing low protein similarity with the different *Bacillus* defective prophages, the regions perform a similar function (bactericidal). These findings are in agreement with the work of [36], who deliberated that in spite of similar genomic organisation, XkdV, a putative tail fibre protein of the PBSX region, differed from the defective prophage PBP180 corresponding operons ORF30, ORF33, and ORF35. Based on the similarities in operon architecture and differences of *Bl* 1821L with the identical regions, it is likely that these regions were acquired from a common distant ancestor.

Bacteria seldom exist in isolation, and generally reside within diverse microbial communities [85]. A fundamental requirement for survival in these complex environments is to gain access to scarce resources, i.e., nutrients and space, by competing with the other inhabitants. The bacteria are not only in competition with distantly related microorganisms, but also with phylogenetically related kin bacteria, which are likely the contenders for the same ecological niche [85,86]. Therefore, competition with the kin bacteria requires highly specific mechanisms to outcompete—but not to kill—individuals from the same clonal population (strains which have identical genomes). Hence, the arsenal employed against kin needs to be highly specialised, with a narrow spectrum of activity [87,88]. The findings of the antibacterial spectrum of putative phage-tail like structures (bacteriocins) of *Bl* 1821L against itself and the kin strains (*Bl* 1951, *Bl* Rsp, *Bl* CCEB 342, *Bl* NRS 590, *Bl* NCIMB) were significant in the context of interbacterial competition. Importantly, the expression of broad-spectrum activity of the *Bl* 1821L putative ABPs may provide an insight into complex relationships within bacterial communities.

## 5. Conclusions

Initially, it was hypothesised that the Tectivirus-like structures might be involved in the loss of virulence of entomopathogenic strain *Bl* 1821L [18]. However, no plausible evidence was found in this work. Overall, this research was the first to identify, purify, and characterise an inducible and bactericidal phage-like element PBSX gene *xkdK* encoded in an entomopathogenic bacterium *Bl* 1821L genome and compare the encoding region to other bacteria, including the closely related strain *Bl* 1951. Despite the broad spectrum of activity of the PEG 8000 precipitated lysate of *Bl* 1821L, the absence of bioactivity against a kin strain *Bl* CCEB 342 can provide an insight into the interbacterial competition at the community level. Furthermore, the decrease in the number of viable cells of the *Bl* 1951 due to the antibacterial activity of identified phage tail-like particles is likely to have implications with regard to harnessing the insecticidal potential of these strains in future.

## Figures and Tables

**Figure 1 biomolecules-12-01154-f001:**
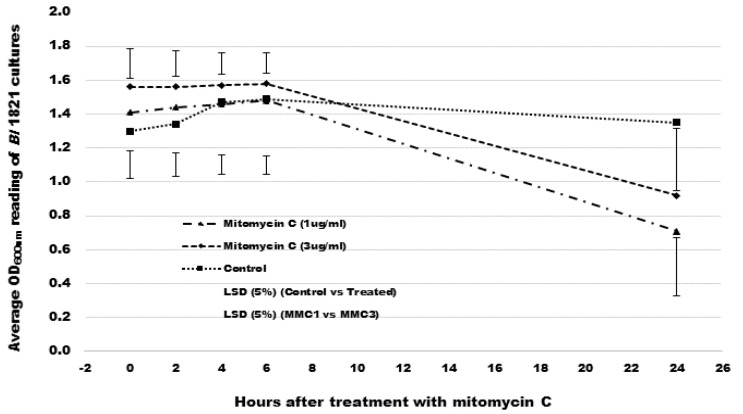
Spectrophotometer reading (OD_600 nm_) reading of *Bl* 1821L cultures after treatment with mitomycin C at 1 µg/mL and 3 µg/mL concentrations and control.

**Figure 2 biomolecules-12-01154-f002:**
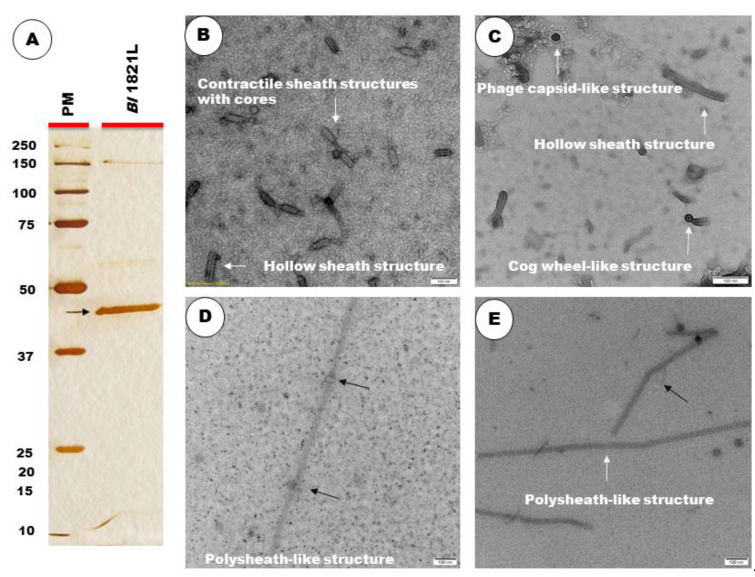
SDS-PAGE of sucrose density gradient purified and 10 kD MWCO membrane concentrated lysate showing a protein band of ~48 kD [(**A**), shown with dark arrow]. Electron micrographs of crude lysate of *Bl* 1821L showing the structures with hollow sheath, contractile sheath with cores, phage capsid-like, and cog wheel-like [(**B**,**C**), shown with white arrow]. Purified lysate showing a hollow tube intermittently joined with knot-like structures [(**D**), shown with dark arrow), and without knot-like structures [(**E**) shown, with dark arrow] to form a polysheath. Scale bar = 100 nm.

**Figure 3 biomolecules-12-01154-f003:**
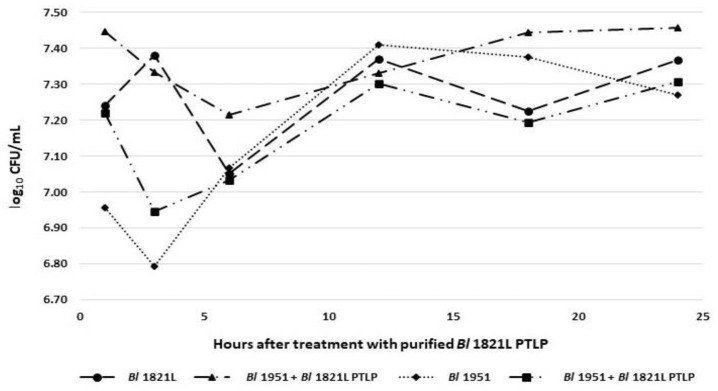
Number of viable cells (log_10_ CFU/mL) of *Bl* 1821L and *Bl* 1951 with/without treatment of purified *Bl* 1821L putative phage tail-like protein (~48 kD) after incubation at 30 °C over 24 h.

**Figure 4 biomolecules-12-01154-f004:**
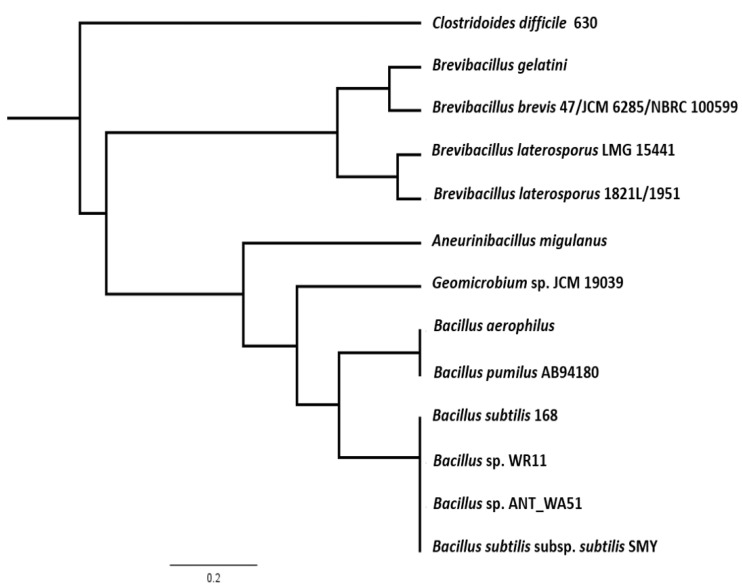
Dendrogram showing alignment *Bl* 1821L and *Bl* 1951 identified (A0A518VEB0) phage-like element PBSX protein XkdK with similar proteins from other Gram-positive bacteria including *Aneurinibacillus migulanus* (A0A0D1WNL8), *Bacillus aerophilus* (A0A410KN98), *Brevibacillus gelatini* (A0A3M8B733), *Brevibacillus laterosporus* LMG 15441 (A0A075R9L5); uncharacterised protein of *Brevibacillus brevis* (strain 47/JCM 6285/NBRC 100599), and similar proteins of other Gram-positive bacteria including *Bacillus* phage PBP180 (R4JQA6), *Bacillus* sp. ANT_WA51 (A0A5B0B6Z4), *Bacillus* sp. WR11 (A0A410QZ71), *Bacillus subtilis* subsp. *subtilis* str. SMY (A0A6H0H1P2), *Bacillus subtilis* 168 (P54331), *Clostridioides difficile* 630 (Q18BN0), and *Geomicrobium* sp. JCM 19039 (A0A061P351) using the Geneious basic. Key is 0.2 nt substitutions per site.

**Figure 5 biomolecules-12-01154-f005:**
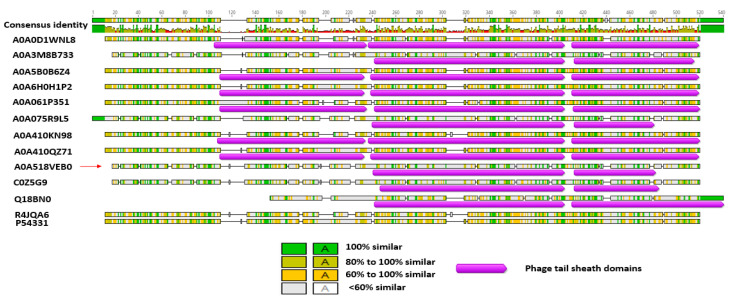
Amino acid alignment and percentage identity of the identified phage-like element PBSX protein XkdK accession A0A518VEB0 (red arrow) of *Bl* 1821L and *Bl* 1951 with the phage tail-sheath proteins of *Aneurinibacillus migulanus* (A0A0D1WNL8), *Bacillus aerophilus* (A0A410KN98), *Brevibacillus gelatini* (A0A3M8B733), *Brevibacillus laterosporus* LMG 15441 (A0A075R9L5); uncharacterised protein of *Brevibacillus brevis* (strain 47/JCM 6285/NBRC 100599) (C0Z5G9), and similar proteins of other Gram-positive bacteria, including *Bacillus* phage PBP180 (R4JQA6), *Bacillus* sp. ANT_WA51 (A0A5B0B6Z4), *Bacillus* sp.WR11 (A0A410QZ71), *Bacillus subtilis* subsp. *subtilis* str. SMY (A0A6H0H1P2), *Bacillus subtilis* 168 (P54331), *Clostridioides difficile* 630 (Q18BN0), and *Geomicrobium* sp. JCM 19039 (A0A061P351) using the Geneious basic.

**Figure 6 biomolecules-12-01154-f006:**
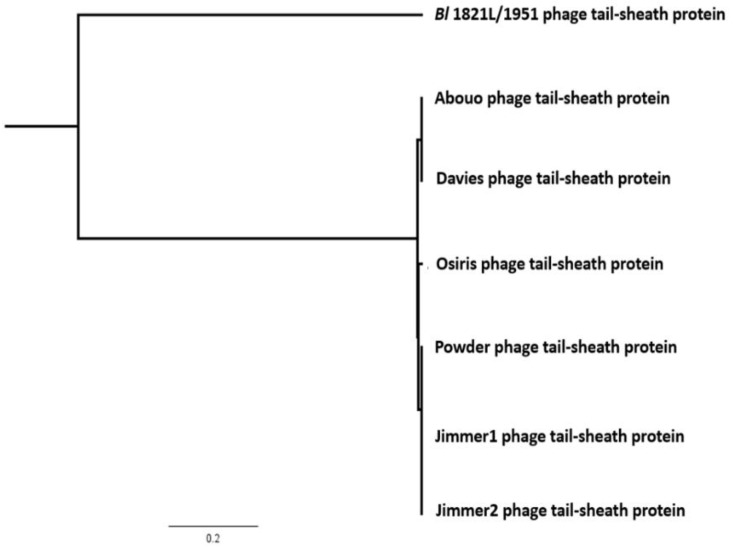
Dendrogram showing the alignment of identified putative phage tail-sheath protein (A0A518VEB0) of *Bl* 1821L and *Bl* 1951 with the similar protein of different *Bl* phages including Abouo (S5MUG6), Davies (S5MCF5), Osiris (A0A0K2CNL4), Powder (A0A0K2FLW7), Jimmer1 (S5MNC1), and Jimmer2, (S5MBG7). Key is 0.2 nt substitutions per site.

**Figure 7 biomolecules-12-01154-f007:**
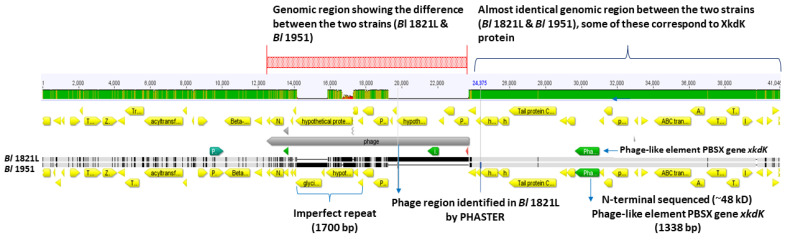
Genome alignment of the *Bl* 1821L and *Bl* 1951 PBSX-like region encoding phage-like element PBSX protein XkdK. Differences between the two strains (shown in red shaded box) with 1700 bp long imperfect repeats of glycine rich proteins residing in the *Bl* 1951 genome and a corresponding putative phage region in the *Bl* 1821L genome. On the top bar, green indicates areas of high homology; olive green indicates areas of lower homology. Grey indicates identical sequences on lower bars; black indicates areas of base differences.

**Figure 8 biomolecules-12-01154-f008:**
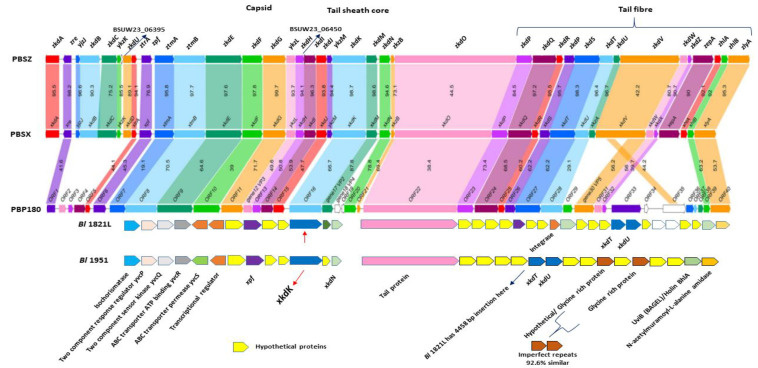
Schematic genomic alignment of identified phage-like element PBSX gene *xkdK* (shown with red arrow) encoding region in the *Bl* 1821L and *Bl* 1951 genomes with defective prophage PBSZ, PBSX, and PBP180 of *B. subtilis* subsp. *spizzenii* W23, *B. subtilis* 168, and *Bacillus* phage PBP180 [36].

**Figure 9 biomolecules-12-01154-f009:**
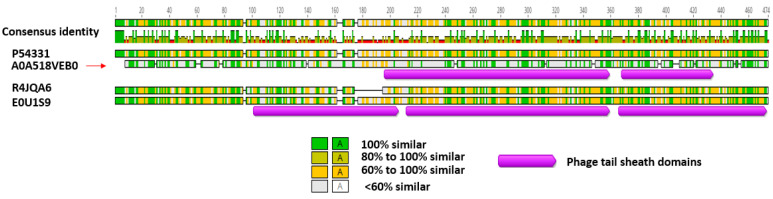
Amino acid alignment and percentage identity of identified phage-like element PBSX protein XkdK accession A0A518VEB0 (shown with red arrow) of *Bl* 1821L and *Bl* 1951 with similar proteins of defective prophages of *Bs* W23 (E0U1S9), *Bs* 168 (P54331), and *Bacillus* phage PBP180 (R4JQA6) using the Geneious basic.

**Table 1 biomolecules-12-01154-t001:** Antibacterial activity of *Bl* 1821L mitomycin C induced culture after PEG8000 precipitation against *Bl* 1821L and *Bl* 1951.

*Bl* 1821L as the Host Bacterium		*Bl* 1951 as the Host Bacterium	
Dilution Level	Zone of Inhibition Diameter (mm)	Dilution Level	Zone of Inhibition Diameter (mm)
FS *	12.5	FS	12.5
10^−1^	– **	10^−1^	10.5
10^−2^	–	10^−2^	13.5
10^−3^	–	10^−3^	10.5
10^−4^	–	10^−4^	–
10^−5^	–	10^−5^	–
10^−6^	–	10^−6^	–
10^−7^	–	10^−7^	–
10^−8^	–	10^−8^	–
Control ***	–	Control	–

* FS = Full strength. ** = No zone of inhibition. *** = For control LB broth pipetted on the lawn of host bacteria.

**Table 2 biomolecules-12-01154-t002:** Antibacterial spectrum of *Bl* 1821L mitomycin C-induced culture cell free supernatant after PEG 8000 precipitation against various Gram-positive bacteria.

Host Bacterium	Host Bacterium Isolate/Strain	Sensitivity to Induced *Bl* 1821L CFS
*Bacillus megaterium*	3-2	– *
*Bacillus megaterium*	S1	–
*Bacillus subtilis*	EM-13 (Tp5)	–
*Brevibacillus laterosporus*	1951	+ **
*Brevibacillus laterosporus*	1821L	–
*Brevibacillus laterosporus*	Rsp	+
*Brevibacillus laterosporus*	CCEB 342	–
*Brevibacillus laterosporus*	NRS 590	+
*Brevibacillus laterosporus*	NCIMB	+
*Carnobacterium maltaromaticum*	3-1	+
*Fictibacillus rigui*	EM-14 (FJAT 46895)	–
*Oceanobacillus* sp.	EM-12 (R-31213)	–
*Oerskovia enterophila*	3-3	–
*Paenibacillus* sp.	15.12.1	–

* – = No zone of inhibition. ** + = Zone of inhibition.

**Table 3 biomolecules-12-01154-t003:** BLASTp comparisons of ~48 kD identified putative phage tail protein of *Bl* 1821L.

Accession	Protein	Host Bacterium	Amino Acids (%) Similarity to the Identified Accession A0A518VEB0
A0A075R9L5	Phage tail sheath	*Brevibacillus laterosporus* LMG 15441	90.3
M8E4N0	Uncharacterised	*Brevibacillus borstelensis* AK1	69.9
A0A3M8DWU9	Phage tail	*Brevibacillus fluminis*	68.0
C0Z5G9	Uncharacterised	*Brevibacillus brevis* (strain 47/JCM 6285/NBRC 100599)	68.1
A0A6M1UCS4	Phage tail	*Brevibacillus* sp. SYP-B805	66.6
A0A0X8D434	Phage tail sheath	*Aneurinibacillus* sp. XH2	49.1
A0A2Z2KS77	Phage tail sheath	*Paenibacillus donghaensis*	48.3

## Data Availability

All data generated or analysed during this study are included in this published manuscript and its Appendix A.

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
