# Peer review of "Isolation, Purification, and Characterisation of a Phage Tail-Like Bacteriocin from the Insect Pathogenic Bacterium Brevibacillus laterosporus"

_biomolecules, 2022, doi:10.3390/biom12081154_

Round 1

Reviewer 1 Report

The authors have put forth an incredibly well-written manuscript on a newly discovered bactericidal phage-like element of the entomopathic bacteria, Brevibacillus. I have now substantive comments. The work details the authors’ thought process and experiments to characterize this protein in detail. It is excellent and for me, highlights how incredibly much work there is to describe phage genes that undoubtedly have important ecological roles in microbial communities. Below are a few minor comments.

Minor comments:

  1. The abstract could be made more accessible. It is quite technical without telling high-level results.

  2. Not sure if it was just my copy, but the figure legend labels seem messed up. Instead of “Figure …..” some of the figures are labeled “Scheme…..”. Figure 3 was incorrectly labeled Figure 1 I think.

  3. Some strange, large spaces in a few spots (e.g., lines 286, 295, 421, 

  4. There was a download link (in supplement) for data hosted on the author’s website. It would be preferable to have this data get a permanent doi. I checked the link. It worked, so that is good.

  5. I’m not sure I completely understand the key on Figure 4 (alignment of PbsX phage-like elements). % identity is a useful metric for comparing two things (how many AAs are different). Is that what is being shown, and if so, what is each genome being compared to? The consensus? It seems to be an alignment, in which case, the conservation plot in the top row is useful. 

  6. Given that Figure 5 is just a dendogram, is there some reason that it cannot be combined with Figure 4? Seems like a marginal plot would work?

  7. Please specify the key units for both dendrograms.

  8. Is Figure 6 a dendogram? I see no evenly spaced nodes.

  9. There are many undescribed features in Figure 7. For example, what does the coloration pattern (green, yellow, red) in the first row mean? What are the black vertical bars? Could also drop gene names that get cut off. Or not. Very minor criticism, but this figure seems overly complicated and busy.

  10. Does the publisher allow for abbreviated (et al.) in-text citations (e.g., line 516)?

Author Response

Thanks

Reviewer 2 Report

In the manuscript entitled “Isolation, purification, and characterisation of a phage tail-like 2 bacteriocin from the insect pathogenic bacterium Brevibacillus laterosporus" authored by Tauseef K. Babar et al., the authors present the results regarding a bacteriocin obtained from a selected bacterium. The authors refer to an important topic in the field, still modifications are required as follows:

Point 1 - Abstract - consider the methods and the conclusion as part of the abstract. The current form details the results, but not the advanced and complex protocol and the methods and also the contribution to the field of knowledge.

Point 2- Introduction - it is too short and it should be developed; the authors could consider a paragraph to better point out the novelty and the relevance of their research given previous studies

Point 3 - Results and discussion - Authors should pay attention and consider the corresponding corrections for

- the figure and tables numbers and legends - lines 210, 225, 230, 249, 267, 284

- subchapters numbers -lines 235, 254

Point 4 - Conclusion - should be modified to better underline the originality and the contribution of the study

Point 5 - the references list contains many outdated studies; the authors are advised to consider only the relevant, fundamental ones.

-

Author Response

Thanks

Reviewer 3 Report

In this study we isolated, purified, and characterised the putative antibacterial phage tail-like structures of the PBSX-like region from an insect pathogenic isolate Bl 1821L and compared the encoding region to other bacteria, including the closely related strain Bl 1951.

In manuscript it is necessary to write °C in this form.

Correct ml as mL.

Abstract and Introduction is very well written.

In material and methods I miss more information about strains described in part 2.1. For example species, resources etc. 

Were there some positive and negative controls used for antimicrobial activity?

How many replicates were evaluated for inhibition zones. Usually it is three times but in the results is SD missing.

Table 1 is ununderstand. Please describe more information about results.

Some of the results are with references. It is necessary to make some relevant changes in discussion.

Conclusion is very general.

The study is very interesting and new, but still it is necessary to do some correction of the manuscript.

Author Response

Thanks

Round 2

Reviewer 2 Report

The authors have addressed all concerns of my previous review and i recommend it for acceptance.